# Fibrinogen and Antifibrinolytic Proteins: Interactions and Future Therapeutics

**DOI:** 10.3390/ijms222212537

**Published:** 2021-11-21

**Authors:** Nikoletta Pechlivani, Katherine J. Kearney, Ramzi A. Ajjan

**Affiliations:** Division of Cardiovascular & Diabetes Research, Leeds Institute of Cardiovascular and Metabolic Medicine (LICAMM), University of Leeds, Leeds LS2 9JT, UK; N.Pechlivani@leeds.ac.uk (N.P.); K.J.Kearney@leeds.ac.uk (K.J.K.)

**Keywords:** fibrinogen, antifibrinolytic proteins, therapeutics, thrombosis, bleeding

## Abstract

Thrombus formation remains a major cause of morbidity and mortality worldwide. Current antiplatelet and anticoagulant therapies have been effective at reducing vascular events, but at the expense of increased bleeding risk. Targeting proteins that interact with fibrinogen and which are involved in hypofibrinolysis represents a more specific approach for the development of effective and safe therapeutic agents. The antifibrinolytic proteins alpha-2 antiplasmin (α2AP), thrombin activatable fibrinolysis inhibitor (TAFI), complement C3 and plasminogen activator inhibitor-2 (PAI-2), can be incorporated into the fibrin clot by FXIIIa and affect fibrinolysis by different mechanisms. Therefore, these antifibrinolytic proteins are attractive targets for the development of novel therapeutics, both for the modulation of thrombosis risk, but also for potentially improving clot instability in bleeding disorders. This review summarises the main properties of fibrinogen-bound antifibrinolytic proteins, their effect on clot lysis and association with thrombotic or bleeding conditions. The role of these proteins in therapeutic strategies targeting the fibrinolytic system for thrombotic diseases or bleeding disorders is also discussed.

## 1. Introduction

The formation of obstructive intravascular thrombi remains a significant cause of morbidity and mortality worldwide [1]. These thrombi can form in arterial and venous vascular beds with the former having a rich presence of platelets [2,3]. This explains why it is mainly antiplatelet therapies that are chosen for the prevention of atherothrombotic disease, while anti-coagulants are used for the treatment and prevention of venous occlusion. However, the clinical management of arterial disease has undergone constant change over the past decade, as clinical outcome studies have shown that the combination of an antiplatelet and an anticoagulant is particularly effective at preventing atherothrombotic events [4,5]. The beneficial effects of combination therapies are not surprising given recent studies demonstrating that fibrin clot characteristics are predictors of clinical outcome in individuals at high risk of atherothrombosis [6,7,8]. Anticoagulants typically reduce fibrin network formation and can also make clots less robust, thus decreasing resistance to lysis, in turn reducing the risk of thrombotic vascular occlusion. A central difficulty in preventing vascular occlusion is the increased risk of bleeding events with more powerful anti-thrombotic agents. Therefore, there is a fine balance between inhibiting platelet function/fibrin network formation and ensuring bleeding risks are kept to a minimum. While newer antiplatelet and anticoagulant therapies are more effective at preventing thrombosis, risk of bleeding remains high. Rather than using powerful agents that have a “global effect” on platelet function and/or coagulation proteins, a more balanced strategy would be to target fibrin clot formation and breakdown, thus having agents with an improved efficacy/safety ratio. One of these potential pathways is targeting the factors responsible for hypofibrinolysis, given this is a known risk factor for thrombosis even with the use of powerful antiplatelet agents [6,7].

Altered incorporation of antifibrinolytic proteins into the fibrin network is an important mechanism that determines fibrinolysis potential [9]. Unlike the clinical use of warfarin (which inhibits synthesis of four coagulation proteins, factors II, VII, IX, and X) [10] or novel oral anticoagulants (NOAC) that inhibit factor IIa or Xa [11], the strategy of interfering with fibrinogen-bound or cross-linked antifibrinolytic proteins will offer the opportunity for a more targeted approach to improve the hypofibrinolytic environment with the real possibility of low risk of bleeding.

In the current narrative review, we discuss the interaction between fibrinogen and fibrin-bound antifibrinolytic proteins, describing their main characteristics and effects on the fibrinolytic process in different disease states. We also explore the role of these proteins as therapeutic targets to reduce thrombosis or bleeding risk, including latest techniques in the field that modulate the function of these proteins.

## 2. Interactions of Fibrinogen with Antifibrinolytic Proteins

Fibrinogen, a soluble glycoprotein with a molecular weight of 340 kDa, consists of two sets of three polypeptide chains (Aα, Ββ, and γ), encoded by three genes *FGA*, *FGB*, and *FGG.* Release of fibrinopeptides A and B from the N-terminal of the Aα and Ββ chains of fibrinogen by thrombin results in the conversion of fibrinogen to fibrin monomers [12,13]. The fibrin monomers polymerise to form fibrin protofibrils, which subsequently assemble to produce a fibrin network [14]. Fibrinogen plays an important role in several pathophysiological processes including thrombogenesis, inflammation, tissue injury, and atherogenesis. Therefore, it interacts with a number of proteins such as Mac-1 and alpha X beta 2 integrins on the surface of leukocytes, glycoprotein IIb-IIIa receptor on the platelet surface, fibronectin, matrix metalloproteinase-2 (MMP-2), and several growth factors including vascular endothelial growth factor (VEGF), basic fibroblast growth factor (bFGF), and insulin-like growth factor-binding protein 3 (IGFBP-3) [15,16,17,18,19].

This review focuses on the proteins that interact with fibrin(ogen) and are involved in the antifibrinolytic process; these are summarized in Figure 1 and Table 1.

### 2.1. Alpha-2 Antiplasmin (α2AP)

α2AP is a ~70 kDa glycoprotein, a member of the serine protease inhibitor (serpin) family, and is mainly produced in the liver, but can also be synthesised by the kidney and brain [20,21,22,23,24]. Human α2AP gene is *SERPINF2,* located on chromosome 17p13.3 and encodes a single-chain protein of 464 amino acid residues with a 27 amino acid residue signal peptide [22,25]. α2AP is the main physiological inhibitor of plasmin and circulates in plasma at a concentration of approximately 70 μg/mL (1 μM) [26,27].

When α2AP and plasmin form a 1:1 stable complex, either in the circulation or on the fibrin surface, plasmin is inhibited [28]. The half-life of α2AP is 2.6 days, but plasmin-antiplasmin (PAP) complexes have a much shorter half-life of approximately 0.5 days [29]. During clot formation, α2AP becomes covalently cross-linked into the fibrin clot by activated factor XIII (FXIIIa) making the clot more resistant to degradation by plasmin [30]. The cross-linking mainly occurs between glutamine residue at position 14 of the α2AP molecule and lysine residue at position 303 of the α chain of fibrin [31], although additional cross-linking sites on fibrinogen have been proposed [32].

#### 2.1.1. Role of α2AP Genetic and Post-Translational Variants

α2AP undergoes both amino terminal (N-terminal) and carboxyl terminal (C-terminal) proteolytic modifications to produce various α2AP isoforms in the circulation. About 30% of circulating α2AP is the native form with a methionine (Met) residue at the N-terminus (Met-α2AP), and the other 70% is N-terminally cleaved by antiplasmin-cleaving enzyme (APCE) between proline (Pro) residue at position 12 and asparagine (Asn) residue at position 13, resulting in the α2AP form with an Asn residue at the N-terminus (Asn-α2AP) [24]. It has been shown that Asn-α2AP is cross-linked to fibrin by FXIIIa 13 times faster than native Met-α2AP, explaining studies that reported superior inhibition of fibrin clot lysis by Asn-α2AP compared with Met-α2AP [30,33,34,35]. It has also been suggested that genetic variation in the *SERPINF2* gene affects N-terminal heterogeneity of α2AP, as the arginine (Arg)-to-tryptophan (Trp) polymorphism at position 6 was shown to influence the rate of α2AP incorporation into fibrin clots. Specifically, Met-α2AP (Arg6) was cleaved about eight times more rapidly than Met-α2AP (Trp6) [36]. Recently, Bronic et al. have shown in a Croatian cohort that individuals with Arg6Trp α2AP CC genotype had an almost 4-fold higher risk of coronary artery disease compared with Arg6Trp α2AP TT genotype [37].

The C-terminal of α2AP is also post-translationally modified, and two forms are present in plasma of which only one can bind plasminogen, referred to as plasminogen binding α2AP (PB-α2AP) while the other form fails to bind plasminogen and is termed non-plasminogen binding α2AP (NPB-α2AP) [38,39]. The liver produces the PB-α2AP form, and this constitutes 65% of circulating α2AP, while NPB-α2AP is formed in the circulation [25,40,41]. The C-terminal of α2AP plays a significant role in the interaction with plasmin(ogen), as this interaction takes place via the lysine (K) residues positioned in the C-terminal of α2AP (K^418^, K^427^, K^434^, K^441^, K^448^, and K^464^) with the lysine binding sites (LBS) in the kringle domains of plasmin(ogen) [42]. Furthermore, α2AP C-terminal contains an arginine–glycine–aspartic acid (RGD) sequence, important for cell recognition and cell adhesion via integrins. Functionally, this RGD sequence may modulate platelet activation, suggesting a dual role for α2AP on both the cellular and protein arms of coagulation [43,44]. However, this is an area that is incompletely understood, and further research is required to fully elucidate the role of the RGD region in α2AP function.

#### 2.1.2. Effects of Congenital and Acquired Deficiency of α2AP

Congenital deficiency of α2AP, an autosomal recessive condition, causes a rare bleeding disorder. Individuals with homozygous α2AP deficiency can exhibit severe bleeding, while individuals with heterozygous deficiency usually have mild bleeding tendencies or may be asymptomatic [45,46]. Congenital deficiency can be either quantitative with decreased protein levels or qualitative with reduced protein function [47]. Acquired deficiency may be seen in patients with various conditions such as acute leukaemia, amyloidosis, and severe liver disease [48,49,50,51]. Reduced levels of α2AP have also been reported in patients with disseminated intravascular coagulation (DIC) and those undergoing thrombolytic therapy [46,52].

#### 2.1.3. Role of α2AP in Thrombotic Disorders

Increased levels of α2AP in man have shown associations with ischaemic stroke [53], while animal work has demonstrated a link between α2AP and venous thrombosis (α2AP^−/−^ mice were protected against thrombosis) [47,54]. Several animal studies also investigated the role of α2AP in pulmonary embolism, ischaemic stroke, thrombotic thrombocytopenic purpura (TTP), and arterial thrombosis, in which α2AP was shown to be involved in the formation and removal of venous thrombi in mice [55,56,57,58,59].

### 2.2. Thrombin Activatable Fibrinolysis Inhibitor (TAFI) 

TAFI is a zinc-dependent metallocarboxypeptidase, synthesised by the liver and megakaryocytes as a propeptide consisting of 423 amino acids; when the 22 amino acid signal peptide is removed, the 56 kDa proenzyme containing 401 amino acids is secreted into the blood circulation [60,61,62]. The gene encoding human TAFI, *CPB2*, was mapped to chromosome 13 (13q14.11) and contains 11 exons [63,64]. The concentration of TAFI in plasma varies from 4 to 15 μg/mL and is also stored within platelet α-granules, at approximately 50 ng per 10^9^ platelets [65,66]. Therefore, TAFI concentration at the site of thrombus formation is much higher than circulating levels. TAFI belongs to and shares structural characteristics with the subfamily A of metallocarboxypeptidases consisting of two domains: the N-terminal activation peptide and a catalytic domain [67].

#### 2.2.1. TAFI Activation and Role of TAFI in Fibrinolysis

TAFI can be activated by thrombin, by plasmin, or by thrombin in complex with thrombomodulin. The latter causes the most efficient TAFI activation followed by plasmin, while thrombin is the weakest activator [68,69]. No physiological inhibitors of TAFI have been identified, and therefore it is speculated that this protein is regulated by other mechanisms that involve intrinsic thermal instability [70]. TAFI is cross-linked by FXIIIa into the fibrin clot via three major amino acid sites, Glutamine (Gln)2, Gln5, and Gln292, and this may also play a role in facilitating its activation, enhancing its activity and protecting the fibrin clot from plasmin degradation [71]. Activated TAFI (TAFIa) protects fibrin clots from lysis by cleaving off C-terminal lysine residues from the fibrin surface which reduces plasminogen and tPA binding, consequently limiting plasmin production [60,72,73,74]. Moreover, TAFIa reduces plasmin binding by removing C-terminal lysine residues from fibrin thus enhancing plasmin and α2AP interactions [75]. The antifibrinolytic activity of TAFI depends on the initial proenzyme plasma concentration, the TAFIa generation rate and its half-life (the half-life of TAFIa is about 10 min at 37 °C, about 40–50 min at 30 °C and about 120–150 min at 22 °C) [76,77].

#### 2.2.2. Role of TAFI Genetic and Post-Translational Variants

Two of nineteen identified single-nucleotide polymorphisms (SNPs) located in the coding region result in amino acid substitutions, which create four TAFI isoforms of which the 325 Thr/Ile polymorphism affects TAFIa stability and antifibrinolytic activity [61,78,79]. The 438 G/A polymorphism was reported to be a risk factor for developing venous thrombosis [80,81]. The association of TAFI single SNPs, 438 G/A, 505 G/A, and 1040 C/T with protein plasma levels, and the risk of deep vein thrombosis (DVT) was investigated. Carriers of 505 G allele showed lower plasma TAFI levels and increased DVT risk compared with 505 A carriers [82]. These contradictory results are intriguing and may be related to alterations in protein function induced by the polymorphism.

Analysis of post-translational modification of human TAFI revealed five N-linked glycans, four of which are attached to the activation peptide and one to the catalytic domain involved in substrate binding. Upon TAFI activation, the activation peptide and attached glycans are removed, causing changes to protein properties including a shift in the isoelectric point and a reduction in solubility [83].

#### 2.2.3. Effects of TAFI Deficiency

TAFI plasma levels are reduced in advanced liver disease, which may contribute to bleeding tendency observed in cirrhosis [84]. Homozygous TAFI-deficient mice develop normally, do not exhibit bleeding tendencies, and plasma clot lysis is not affected [85]. Moreover, in vivo animal work has shown that TAFI deficiency does not affect rate of arterial or venous thrombus formation, and has no effect on survival rate following experimental vascular occlusion [85]. However, another study has reported enhanced fibrinolysis in TAFI deficient mice and decreased accumulation of fibrin in the lungs in a batroxobin-induced pulmonary embolism model [86]. Furthermore, TAFI deficient mice demonstrated a decrease in thrombus size in FeCl_3_-induced vascular injury models and enhanced fibrinolysis in a thromboembolism model [87,88]. More recently, it was reported that functional TAFI deficiency in haemophilia promotes maladaptive vascular remodelling in the joints after bleeding [89], suggesting a more diverse role for this protein.

#### 2.2.4. Role of TAFI in Thrombotic Disorders

In the Leiden Thrombophilia Study (LETS), elevated TAFI plasma levels were associated with a small increase in the risk of venous thrombosis [90]. Furthermore, patients with high TAFI and high levels of one of the factors VIII, IX, or XI had higher relative risk for recurrence compared with patients with low levels of TAFI and one of these factors [91]. In addition to venous thrombosis, elevated TAFI levels have shown associations with ischaemic stroke [92]. In contrast, another study reported that patients with a recent myocardial infarction had lower TAFI levels and that high TAFI levels were associated with reduced risk of myocardial infarction [93]. As alluded to earlier, it is possible that altered protein function accounts for the association between lower levels and disease, and further mechanistic studies in this area are required.

### 2.3. Complement C3

Complement protein C3 is a main component of the human complement system and plays an important role in innate immunity. C3 is a 187 kDa protein consisting of two chains (alpha and beta) and belongs to the α2-macroglobulin family [94,95]. The gene encoding human C3, called *C3*, is located in chromosome 19 (19p13.3) and contains 41 exons [96,97]. C3 is synthesised mainly by the liver but is also produced by immune cells and is present in plasma at high concentrations of approximately 1.2 mg/mL [98,99]. Evidence suggests a link between the complement system and the coagulation/fibrinolysis cascade [100].

#### 2.3.1. Interaction of C3 with Fibrin(ogen) and Role in Fibrinolysis

C3 has been identified as a novel clot component that is able to bind to immobilised fibrinogen and fibrin with high affinity [101,102]. C3 incorporation into fibrin clots results in prolongation of fibrinolysis in a concentration-dependent manner, while binding data from the same study indicated that there are two high-affinity binding sites for C3 on both fibrinogen and fibrin [101,102]. We have recently shown, using microarray analysis, that the Bβ chain of fibrinogen contains key binding sites for C3 [103]. C3 can be both bound (non-covalently), and cross-linked to fibrin networks by FXIIIa [104,105]. Plasma levels of C3 are independently associated with a history of acute or chronic vascular injury [106,107,108,109]. Moreover, C3 has shown an association with fibrin clot lysis in 837 type 2 diabetes individuals, while increased incorporation into fibrin networks of type 1 diabetes individuals suggests this protein may become a diabetes-specific target to improve hypofibrinolysis [110,111].

#### 2.3.2. Role of C3 Genetic and Post-Translational Variants

C3 genetic variants have been linked with various conditions such as severe pre-eclampsia, systemic lupus erythematosus (SLE), and advanced age-related macular degeneration [112,113,114,115]. However, links with vascular disease are less convincing; the Arg102Gly polymorphism has shown associations with C3 plasma levels, but not with the presence or severity of coronary artery disease, casting doubts on the importance of this polymorphism in vascular pathology [116].

Post-translational modifications of C3, and in particular glycation, may also affect its properties. Glycation of C3 may cause changes in its structure, affecting the immune properties of C3 [117]. We have recently shown that glycation of C3 enhances the antifibrinolytic activity, although the exact glycated residues are yet to be identified [103].

#### 2.3.3. Effects of C3 Deficiency

Deficiencies in proteins of the complement system are usually hereditary and associated with increased susceptibility to infections [118]. Recent animal studies have revealed an association between C3 deficiency and increased angiogenesis, which may have implications for vascular occlusive disease [119,120].

#### 2.3.4. Role of C3 in Thrombotic Disorders

A few studies investigated the relationship between C3, SLE, and thrombosis. Early work has shown an association between C3 plasma levels and coronary artery disease [106]. Increased C3 levels have also been documented in pregnancy-related venous thrombosis [121]. However, low levels of C3 in SLE patients were associated with increased risk of thrombosis [122]. It is not only quantitative changes in the protein that are associated with thrombosis; qualitative changes can also show associations with thrombosis (such as an increase in protein phosphate content) [123].

### 2.4. PAI-2

Human plasminogen activator inhibitor-2 (PAI-2) is a single chain protein of 415 amino acids, primarily found as a 47 kDa non-glycosylated intracellular form, however it is also secreted as a 60 kDa glycosylated protein [124]. Expression of PAI-2 has been detected in monocytes, macrophages, keratinocytes, fibroblasts, and the placenta [125,126]. The gene encoding human PAI-2, *SERPINB2,* is located on chromosome 18 (18q21-23) and consists of eight exons [127]. PAI-2 is a member of the serpin superfamily and was identified as a placental tissue-derived urokinase-type plasminogen activator (uPA) inhibitor and, to a lesser extent, a tPA inhibitor [126,128,129,130]. Plasma concentrations of PAI-2 are normally below detection limit, however during pregnancy, elevated levels have been reported, which also applies to some serious conditions such as myelomonocytic leukaemias and severe sepsis [126,130,131,132].

#### 2.4.1. Cross-Linking to Fibrin and Inhibition of Plasmin Generation by PAI-2

Unlike other serpins, PAI-2 has an extension of exon 3 that encodes a unique domain named C-D loop [133]. This contains glutamine residues that form a substrate for transglutaminases and FXIIIa [134]. It has been shown that Gln83 and 86 residues are important for cross-linking PAI-2 to several Lys residues (148, 176, 183, 230, 413, and 457) on the fibrin α chain [135,136]. Cross-linking to α Lys 148 may be crucial for PAI-2 activity, given this site is close to a tPA binding site (148–160) [137]. Furthermore, cross-linking of PAI-2 and α2AP to fibrin α chain can occur simultaneously using different lysine residues, further enhancing resistance to fibrin clot lysis [138].

#### 2.4.2. Role of PAI-2 Genetic and Post-Translational Variants

Two PAI-2 variants (variant A consisting of Asn120, Asn404, and Ser413, and variant B consisting of Asp120, Lys404, and Cys413) have shown no association with myocardial infarction [139], although others documented an association [140]. Furthermore, the PAI-2 variant rs8093048 was associated with coronary artery disease in Chinese Han population (a total of 925 individuals participated in this study: 407 patients with coronary artery disease and 518 healthy controls) [141], while Ser(413)/Cys (rs6104) failed to show an association with premature coronary artery disease in a smaller study of southern Iran population (200 patients and 200 control subjects participated in this study) [142].

#### 2.4.3. Effects of PAI-2 Deficiency

Deficiency of PAI-2 has not been reported in humans or other mammals, suggesting a vital role of this protein in embryogenesis. However, PAI-2 deficient mice had normal development, survival, fertility, and response to infections [143]. While no clear links with vascular disease have been documented, PAI-2 deficiency has shown an association with malignant tumour growth and metastasis by mechanisms that remain unclear [144].

#### 2.4.4. Role of PAI-2 in Thrombotic Disorders

While PAI-2 has been investigated in malignant disorders, its role in vascular disease in the absence of malignancy remains unclear. An association of PAI-2 with DVT was reported in animal model of stasis, while PAI-2 deficient mice showed enhanced venous thrombus resolution [145]. This suggests a role for PAI-2 in venous thrombosis and human studies to investigate the role of this protein in clinical thrombosis are warranted.

## 3. Targeting the Antifibrinolytic Proteins for Developing Therapeutics

### 3.1. Therapeutics for Thrombotic Disorders

Antifibrinolytic proteins represent attractive targets for the development of therapeutics to modulate thrombosis risk. Various methodologies have been explored to inhibit the functions of antifibrinolytic proteins; most revolve around the production of protein-specific monoclonal antibodies.

#### 3.1.1. Targeting α2AP

Monoclonal antibodies that inhibit α2AP have been shown to enhance fibrinolysis [146,147,148,149]. A monoclonal antibody which inactivated α2AP was shown to reduce brain infarction, swelling, and haemorrhage in a murine model of thromboembolic stroke [150]. Furthermore, an α2AP inactivating antibody increased thrombus dissolution and reduced stroke mortality compared to tPA therapy in mice [59]. More recently, Singh and colleagues used a humanised α2AP mouse model of pulmonary embolism to investigate the effects of pharmacological recombinant tPA (r-tPA) and α2AP inhibition on fibrinolysis and bleeding [151]. α2AP-inactivating monoclonal antibody alone or combined with low dose r-tPA enhanced thrombus dissolution with low bleeding risk [151]. The α2AP-inactivating monoclonal antibody was also demonstrated to increase fibrinolysis in a mouse model of venous thrombosis, suggesting a role for α2AP inhibition in DVT [54].

Alternative approaches to antibodies have also been explored; synthetic peptides mimicking the N-terminal of α2AP were used as “competitive substrates”, thus reducing FXIIIa-mediated α2AP incorporation into fibrin networks [152,153]. Fusion of human serum albumin (HSA) to the α2AP N-terminal motif has also been shown to reduce fibrinolytic resistance by a similar mechanism [154]. The α2AP-HSA protein could be cross-linked by FXIIIa to fibrinogen and fibrin, competing with native α2AP and reducing the α2AP-dependent resistance to fibrinolysis of plasma clots [154]. Furthermore, microplasmin, which is the shortened version of plasmin containing only the catalytic domain, was reported to neutralize α2AP activity in healthy volunteers but the development of microplasmin as a therapeutic for cardiovascular disease has not progressed any further [155]. Moreover, given the difference in antifibrinolytic efficacy of Met-α2AP and Asn-α2AP, APCE inhibitors were developed, which may have a role as mild enhancers of fibrinolysis [35].

#### 3.1.2. Targeting TAFI

Recently, a TAFIa inhibitor, a low molecular weight compound named S62798, has been shown to enhance clot lysis in thromboelastometry experiments using whole blood and also decreased pulmonary fibrin deposition in a mouse in vivo model of thromboembolism [156]. This molecule was also investigated for its effect on bleeding using a rat tail bleeding model; administration of r-tPA was used as a positive control and TAFIa inhibitor was found to be associated with a low risk of bleeding.

Other small-molecule TAFIa inhibitors were assessed in animal studies, such as the Potato Tuber Carboxypeptidase Inhibitor (PTCI), a 39 amino acid peptide isolated from the potato tuber, reported to enhance tPA-induced arterial thrombolysis in rabbits [157]. Another TAFIa inhibitor is the product named BX 528, demonstrated to improve thrombolysis without increasing bleeding in rats, dogs, and rabbits [158]. Most recently, a low-molecular weight oral TAFIa inhibitor, DS-1040, was suggested as a potential therapeutic agent to enhance fibrinolysis with low bleeding risk [159]. Interestingly, DS-1040 has been further evaluated in a phase 1 human study in order to assess safety (including bleeding time), tolerability, pharmacokinetics and pharmacodynamics [160].

In addition to small molecule inhibitors, monoclonal antibodies against TAFI have been raised which can interfere with TAFIa activity or TAFI activation [161]. However, some of these monoclonal antibodies against human TAFI did not have cross-reactivity with mouse or rat TAFI and therefore could not be further tested in animal studies [161]. Five monoclonal antibodies against rat TAFI shown to enhance clot lysis have undergone mechanistic studies, demonstrating the ability to destabilize TAFIa, block access to the protein’s active site, or prevent binding of TAFIa to the fibrin clot [162]. Data from another study of TAFI inhibition by a monoclonal antibody suggested that the antibody MA-TCK26D6 exerts its activity by blocking the access of TAFI activators such as plasmin and thrombin [163]. The MA-TCK26D6 inhibitory monoclonal antibody showed a profound effect in enhancing fibrinolysis in a mouse thromboembolism model.

Nanobodies were also screened against mouse TAFI and one nanobody (VHH-mTAFI-i49) was demonstrated to decrease fibrin deposition in a mouse thromboembolism model [164].

Others have investigated inhibition of two pathways, using a heterodimer diabody against TAFI and PAI-1 in mouse models of thrombosis and stroke. The bispecific antibody was able to exhibit a profibrinolytic effect with low bleeding risk [165].

#### 3.1.3. Targeting Complement C3

C3 prolongs clot lysis, an effect that is exaggerated in individuals with diabetes, making it a potential disease-specific target. Using Affimer technology, we have recently shown that fibrinogen-binding, C3 specific Affimers can modulate clot lysis in plasma samples from healthy controls and individuals with diabetes [166]. These results suggest that C3 represents a promising therapeutic target for the reduction in thrombotic risk and future in vivo animal studies are warranted.

As coronavirus disease (COVID-19) is continuing to spread around the world, C3 inhibition by a C3-targeted drug candidate, AMY-101, was recently evaluated in small independent cohorts of severe COVID-19 patients, given that targeting complement represents one approach for improving COVID-19-mediated immunothrombosis [167,168]. AMY-101 is a third-generation Cp40-based compstatin analogue able to inhibit C3 activation by C3 convertases that is currently in Phase II/III development [169].

Furthermore, research in C3 inhibition by pegcetacoplan (APL-2), a PEGylated C3 inhibitor, is ongoing in patients with paroxysmal nocturnal haemoglobinuria (PNH), a rare acquired life-threatening hematologic disease that causes complications through both haemolysis and thrombosis [170,171].

#### 3.1.4. Targeting PAI-2

Although PAI-2 deficiency has been associated with increased venous thrombus resolution as discussed above, there seems to be low interest in using PAI-2 for the development of new therapeutics. This may be due to the limited and sometimes contradictory evidence linking PAI-2 to increased thrombosis risk.

### 3.2. Therapeutics for Bleeding Disorders

Treatment for bleeding disorders such as haemophilia has primarily focused on replacing the missing coagulation factor. Although recombinant bioengineering has improved various aspects of replacement therapies (i.e., decreased immunogenicity, increased efficacy, and extended half-lives), novel molecules are in development to further improve management of haemophilia [172].

One approach is the use of adjunctive antifibrinolytic therapies to reduce bleeding complications in haemophilia. The addition of TAFI helped to reduce clot lysis in haemophilic plasma and stabilised the fibrin network [173]. This strategy was also effective when TAFI (or thrombomodulin) was added to the plasma of haemophilia patients with FVIII inhibitory antibodies [173]. Another study showed that soluble thrombomodulin (Solulin) improved clot stability in severe haemophilia A by promoting TAFI activation [174]. In particular, the use of low Solulin concentrations prolonged clot lysis by a TAFIa-dependent mechanism [174].

The synthetic lysine analogues tranexamic acid (TXA) and epsilon aminocaproic acid (EACA) interfere with fibrin-plasmin(ogen) interaction, therefore preventing clot lysis, and are widely used clinically as antifibrinolytic agents [13]. Fibrin sealants, consisting of a mix of proteins including fibrinogen, thrombin, FXIII, and antifibrinolytic agents, are also used in surgical procedures, however a number of limitations have been reported to be associated with their application, including increased risk of thrombosis [13]. Engineered haemostatic polymer (PolySTAT) composed of various fibrin-specific binding domains has been shown to enhance clot formation and increase resistance to lysis [175]. Our group has recently demonstrated the potential use of a fibrinogen-binding Affimer protein that provides a novel methodology for stabilizing the fibrin clot and reducing bleeding risk [176]. The fibrinogen-specific Affimer prolonged fibrinolysis across plasma samples from healthy subjects and plasma deficient in FVIII (haemophilia A). Importantly, the addition of Affimer normalised the short lysis time in this condition, indicating that the Affimer has the capability to work in pathological conditions [176]. Given that Affimers can also facilitate fibrinolysis, as discussed above, the Affimer technology has the potential to be used both for thrombotic and bleeding conditions and animal in vivo studies with these molecules are warranted.

Future therapeutic strategies for the treatment of bleeding disorders such as haemophilia A and B are likely to focus on extended half-life coagulation factors, although adjunctive therapies targeting the antifibrinolytic proteins have the potential to improve efficacy of the replaced clotting factors and this remains an area for future research.

A schematic representation of the therapeutic approaches involving antifibrinolytic proteins targeting hypofibrinolysis or clot instability for thrombotic or bleeding disorders, respectively, is shown in Figure 2.

## 4. Conclusions and the Future

While a large number of studies have investigated the role of fibrin-incorporated antifibrinolytic proteins in health and disease, characterization of their exact role in vascular occlusive disease is incompletely understood. This is likely related to the heterogeneity of the population studied, small numbers analysed, and/or the sensitivity of the methodologies applied. While more research in this area is needed, some of these antifibrinolytic proteins are emerging as potential therapeutic targets given their role in disease states and consistent effect on fibrinolysis. Perhaps the antifibrinolytic protein with the most evidence for use as a therapeutic target is α2AP; indeed, several approaches have been explored to modulate protein activity. Monoclonal antibodies against α2AP have been particularly effective at altering α2AP activity, and these have even been tested in phase I and phase II clinical studies, but none have made it into routine clinical practice to date. ΤAFI inhibitors have also been developed, with some showing early promising results, while there has been little investment in developing PAI-2 inhibitors given their inconsistent role in disease.

Interestingly, the use of antifibrinolytic proteins as therapeutic targets is not limited to thrombotic conditions, but also bleeding disorders, where they may prove to be effective as adjunctive therapies or even as main agents to stop blood loss.

Taken together, the molecular mechanisms involved in function of fibrin-incorporated antifibrinolytic proteins are largely understood, but more work is needed to fully elucidate the groups, or subgroups, of individuals who would benefit the most from antifibrinolytic-based therapies. While a number of approaches for modulating the function of antifibrinolytic proteins have been developed, more work is required to ensure that such therapies are effective in vivo (i.e., good efficacy/safety profile) and do not have unwanted “off target” effects. Overall, current evidence suggests that antifibrinolytic-directed therapies have the potential to be novel antithrombotic agents with a low risk of bleeding, while also being relevant to the discovery of agents that can be used in bleeding disorders. Appropriate collaborations between scientists, clinicians, and the pharmaceutical industry should help to make antifibrinolytic-directed therapies part of daily clinical practice.

## Figures and Tables

**Figure 1 ijms-22-12537-f001:**
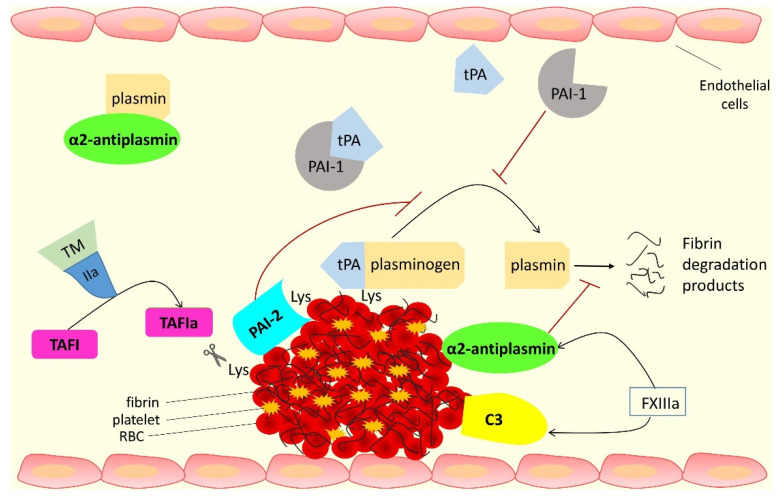
The role of antifibrinolytic proteins in the fibrinolytic process. The fibrin component of the thrombus is degraded by plasmin, generated by tissue plasminogen activator (tPA) activation of plasminogen. Anti-fibrinolytic protein plasminogen activator inhibitor-1 (PAI-1) binds tPA, preventing plasminogen activation. Alpha-2 antiplasmin (α2AP) forms a stable complex with plasmin in the circulation or becomes cross-linked into the fibrin clot by activated FXIII (FXIIIa), which makes the clot more resistant to fibrinolysis. Thrombin activatable fibrinolysis inhibitor (TAFI) is activated by thrombin (IIa) in complex with thrombomodulin (TM). Activated TAFI (TAFIa) cleaves off lysine residues (Lys) from the fibrin surface therefore decreasing plasminogen and tPA binding, thus reducing plasmin generation. TAFI, as well as plasminogen activator inhibitor-2 (PAI-2), can also be cross-linked into the fibrin clot by FXIIIa. Complement C3 is bound and cross-linked to the fibrin clot by FXIIIa, causing prolongation of fibrinolysis.

**Figure 2 ijms-22-12537-f002:**
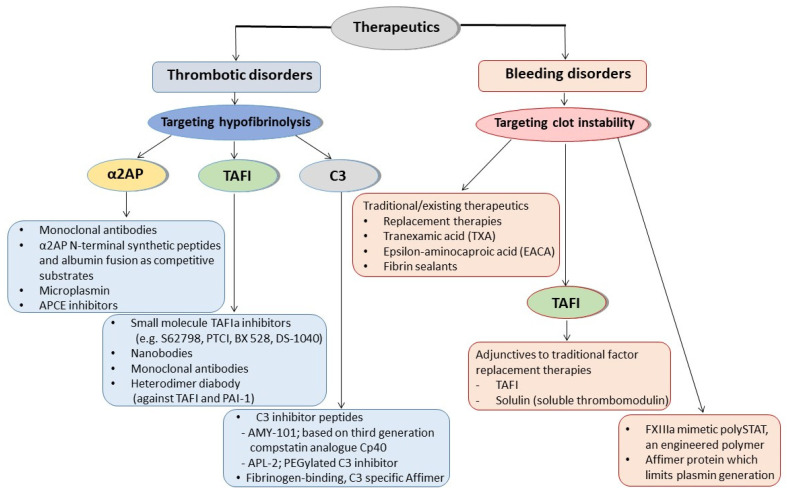
Antifibrinolytic proteins as therapeutic targets for thrombotic or bleeding disorders. Alpha-2 antiplasmin (α2AP), thrombin activatable fibrinolysis inhibitor (TAFI) and complement protein C3 are proteins with antifibrinolytic activity that are promising therapeutic targets in thrombotic disorders. Monoclonal antibodies that inhibit α2AP, synthetic peptides, or albumin fusion proteins that mimic the N-terminus of α2AP and act as ‘competitive substrates’, microplasmin, and antiplasmin-cleaving enzyme (APCE) inhibitors that enhance fibrinolysis have been developed and studied as potential therapeutics to reduce thrombosis. Small molecule TAFIa inhibitors, nanobodies, and monoclonal antibodies against TAFI, as well as heterodimer diabody against TAFI and PAI-1 have also been investigated. Complement C3 inhibitor peptides have been tested as drug candidates for thrombotic diseases, and Affimer technology was employed to develop fibrinogen-binding C3 specific Affimers that modulate clot lysis. On the other hand, antifibrinolytic proteins and processes contributing to clot stability are targeted in bleeding disorders. Traditionally used therapeutics for bleeding disorders mainly focus on replacement of the missing coagulation factor(s). Antifibrinolytic lysine analogues tranexamic acid (TXA) and epsilon-aminocaproic acid (EACA) are also used to limit bleeding. Fibrin sealants are used during surgical procedures to aid the maintenance of haemostasis. TAFI has been investigated as an adjunctive therapy for individuals with bleeding disorders, as has Solulin, a recombinant soluble analogue of human thrombomodulin. More recently, PolySTAT, an engineered protein scaffold that binds fibrin and acts similarly to FXIIIa, was able to strengthen clots and reduce bleeding in animal models. Fibrinogen-specific Affimers have also been developed and shown recently to represent a novel methodology for reducing bleeding.

**Table 1 ijms-22-12537-t001:** Summary of fibrin-bound antifibrinolytic proteins alpha-2 antiplasmin (α2AP), thrombin activatable fibrinolysis inhibitor (TAFI), complement C3, and plasminogen activator inhibitor-2 (PAI-2).

	α2AP	TAFI	C3	PAI-2
Mass (kDa)	~70	56	187	47
Human gene	*SERPINF2*	*CPB2*	*C3*	*SERPINB2*
Synthesis/expression	Liver, kidney, and brain	Liver and megakaryocytes	Liver and immune cells	Monocytes, macrophages, keratinocytes, fibroblasts, and placenta
Circulating plasma concentration	70 µg/mL	4–15 µg/mL	1.2 mg/mL	Below detection limit
Antifibrinolytic function	Direct binding to, and inhibition of, plasmin and cross-linking into the clot making it more resistant to lysis	Protects the clot from lysis by cleaving off C-terminal lysine residues from fibrin, which reduces plasminogen and tPA binding and subsequent plasmin generation	Incorporation into the fibrin clot causes prolongation of fibrinolysis	Cross-linking into fibrin at a site close to tPA binding site affects fibrin clot lysis

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
