# Peer review of "Fibrinogen and Antifibrinolytic Proteins: Interactions and Future Therapeutics"

_ijms, 2021, doi:10.3390/ijms222212537_

Round 1

Reviewer 1 Report

The review of Pechlivani et al. provides an overview of the present knowledge on fibrinogen. The manuscript focuses on fibrinolysis and its regulation by antifibrinolytic proteins. The authors address specifically the pharmacological aspects of the biochemistry of fibrinogen. Further, they describe how regulatory proteins are targeted in antithrombotic therapy. It is an important topic, well written, and updated. Figures help to convey the message and the reference list, quite long, is very thorough. In their conclusion section, the authors present their ideas on the future development of the field. In this way, they provide a balanced commentary.
I have some specific aspects that the authors could consider for revision/clarification.
A short introduction to the structure of fibrinogen and fibrin could be helpful.
Lines 34-37; “The beneficial effects of combination therapies are not surprising given 34 recent studies demonstrating that fibrin clot characteristics, particularly impaired fibrin clot lysis, are predictors of vascular outcome in individuals at high risk of atherothrombosis.” Here the authors refer to the combination of antiplatelet and anticoagulant treatments. Clot lysis, related to fibrinolysis, does not seem to be a key factor in the success of this combination of drugs. 
Lines 44-46; “One of these potential pathways is targeting the factors responsible for hypofibrinolysis, given this is a known risk factor for thrombosis even with the use of powerful antiplatelet agents.” 
Here, the authors seem to be referring to targeting the regulators of the fibrinolytic cascade, and not to coagulation. 
Line 133-143; This text seems to be the figure 1 caption, as it is repeated in the text. The same is the case for lines 461-483 and figure 2.
Line 172; Separate figures and units (i.e. 37 oC).
Figure 2 has very small letters difficult to read.

Author Response

Reviewer 1.

  1. A short introduction to the structure of fibrinogen and fibrin could be helpful.

We thank the reviewer for the helpful suggestion. We have now included a paragraph about the structure of fibrinogen and fibrin (start of section ‘’2. Interactions of fibrinogen with antifibrinolytic proteins’’). In the same paragraph we have also added a few lines about interactions of fibrinogen with other proteins that do not influence fibrinolysis as suggested by reviewer 2.

‘’Fibrinogen, a soluble glycoprotein with a molecular weight of 340 kDa, consists of two sets of three polypeptide chains (Aα, Ββ, and γ), encoded by three genes FGA, FGB and FGG. Release of fibrinopeptides A and B from the N-terminal of the Aα and Ββ chains of fibrinogen by thrombin results in the conversion of fibrinogen to fibrin monomers [12, 13]. The fibrin monomers polymerise to form fibrin protofibrils, which subsequently assemble to produce a fibrin network [14]. Fibrinogen plays an important role in several pathophysiological processes; including thrombogenesis, inflammation, tissue injury and atherogenesis. Therefore, it interacts with a number of proteins such as Mac-1 and alpha X beta 2 integrins on the surface of leukocytes, glycoprotein IIb-IIIa receptor on the platelet surface, fibronectin, matrix metalloproteinase-2 (MMP-2) and several growth factors including vascular endothelial growth factor (VEGF), basic fibroblast growth factor (bFGF) and insulin-like growth factor-binding protein 3 (IGFBP-3) [15-19].’’

  1. Lines 34-37; “The beneficial effects of combination therapies are not surprising given 34 recent studies demonstrating that fibrin clot characteristics, particularly impaired fibrin clot lysis, are predictors of vascular outcome in individuals at high risk of atherothrombosis.” Here the authors refer to the combination of antiplatelet and anticoagulant treatments. Clot lysis, related to fibrinolysis, does not seem to be a key factor in the success of this combination of drugs. 

Thank you for raising this important point. It can be argued that anticoagulant therapies have a dual role: they prevent fibrin clot formation but can also facilitate fibrinolysis by making the clot less robust. However, we take the comment of the reviewer on board and modified the sentence to read: ‘’The beneficial effects of combination therapies are not surprising given recent studies demonstrating that fibrin clot characteristics are predictors of clinical outcome in individuals at high risk of atherothrombosis [6-8]. Anticoagulants typically reduce fibrin network formation and can also make clots less robust thus decreasing resistance to lysis, in turn reducing the risk of thrombotic vascular occlusion.”

  1. Lines 44-46; “One of these potential pathways is targeting the factors responsible for hypofibrinolysis, given this is a known risk factor for thrombosis even with the use of powerful antiplatelet agents.” 
    Here, the authors seem to be referring to targeting the regulators of the fibrinolytic cascade, and not to coagulation. 

We thank the reviewer for this and agree that the way we have written this is not clear. We have edited it to avoid confusion.

‘’ Rather than using powerful agents that have a “global effect” on platelet function and/or coagulation proteins, a more balanced strategy would be to target fibrin clot formation and breakdown, thus having agents with an improved efficacy/safety ratio. One of these potential pathways is targeting the factors responsible for hypofibrinolysis, given this is a known risk factor for thrombosis even with the use of powerful antiplatelet agents.’’

  1. Line 133-143; This text seems to be the figure 1 caption, as it is repeated in the text. The same is the case for lines 461-483 and figure 2.

We thank the reviewer for this comment and we have now changed the format and position of the figure legend text so that it does not look like part of the main text.

  1. Line 172; Separate figures and units (i.e. 37 oC).

Figures are separated from the units now as requested.

  1. Figure 2 has very small letters difficult to read.

We thank the reviewer for the suggestion. Font size of the small letters has been changed from 11 to 14 to make the text in the boxes in Figure 2 easier to read.

Reviewer 2 Report

The review by Pechlivani N. et. al., summarizing the different proteins that interact with fibrinogen and are involved in fibrinolysis, represents a more specific approach for the development of therapeutic agents. Regulation of fibrin-bound alpha-2 antiplasmin (α2AP), thrombin, activatable fibrinolysis inhibitor (TAFI), complement C3 and plasminogen activator inhibitor-2 (PAI-2), affect fibrinolysis differently and may be useful in modulation of thrombosis risk but also has the potential to improve clot stability in bleeding disorders, therefore could be attractive strategy for development of  effective therapeutic agents.

This is a well-written review and discussed the latest technologies that modulate the function of these anti-fibrinolytic proteins in different disease states.

Adding few lines about biochemical properties of fibrinogen/fibrin and their potential as a therapeutic target for thrombosis or bleeding disorders and about other fibrin(ogen) interacting proteins, like MMPs and growth factors, which may or may not influence the fibrinolysis, will strengthen the review.

Author Response

Reviewer 2.

  1. Adding few lines about biochemical properties of fibrinogen/fibrin and their potential as a therapeutic target for thrombosis or bleeding disorders and about other fibrin(ogen) interacting proteins, like MMPs and growth factors, which may or may not influence the fibrinolysis, will strengthen the review.

We thank the reviewer for the helpful suggestion and we agree that it is important to add a few lines about biochemical properties of fibrinogen/fibrin. Please see our response to reviewer 1.

We also appreciate the suggestion of the reviewer for discussing the potential of fibrin(ogen) as therapeutic target for thrombosis or bleeding disorders, however, we feel that this has been extensively discussed in a recent review by our group published in IJMS:

Fibrin(ogen) as a Therapeutic Target: Opportunities and Challenges

Thembaninkosi G. Gaule and Ramzi A. Ajjan

10.3390/ijms22136916